# Memory-Efficient Acceleration of Block Low-Rank Foundation Models on Resource Constrained GPUs

## Abstract

Recent advances in transformer-based foundation models have made them the default choice for many tasks, but their rapidly growing size makes fitting a full model on a single GPU increasingly difficult and their computational cost prohibitive. Block low-rank (BLR) compression techniques address this challenge by learning compact representations of weight matrices. While traditional low-rank (LR) methods often incur sharp accuracy drops, BLR approaches such as Monarch and BLAST can better capture the underlying structure, thus preserving accuracy while reducing computations and memory footprints. In this work, we use roofline analysis to show that, although BLR methods achieve theoretical savings and practical speedups for single-token inference, multi-token inference often becomes memory-bound in practice, increasing latency despite compiler-level optimizations in PyTorch. To address this, we introduce custom Triton kernels with partial fusion and memory layout optimizations for both Monarch and BLAST. On memory-constrained NVIDIA GPUs such as Jetson Orin Nano and A40, our kernels deliver up to $3.76\times$ speedups and $3\times$ model size compression over PyTorch dense baselines using CUDA backend and compiler-level optimizations, while supporting various models including Llama-7/1B, GPT2-S, DiT-XL/2, and ViT-B. Our implementation will be made public upon acceptance.

## 1 Introduction

Large-scale transformer-based foundation models have achieved remarkable success across language understanding, image classification, and generative tasks (Dosovitskiy et al., 2021; Touvron et al., 2023; Brown et al., 2020; Hoffmann et al., 2022; Peebles & Xie, 2022). However, their rapid growth in size is increasingly outpacing the capacity of available hardware. For example, Llama-70B (Touvron et al., 2023) requires over 140 GB of memory simply to load its weights in half-precision format, yet some of today's most powerful commercial GPUs provide only 80 GB. Beyond sheer memory constraints, the reliance of transformer models with dense matrix multiplications introduces significant computational and memory bandwidth bottlenecks during inference. These challenges limit the deployment of models at scale and also their accessibility on resource-constrained devices.

A widely adopted strategy to address these bottlenecks is to approximate weight matrices using low-rank factorizations (Huh et al., 2021; Yaras et al., 2023; Kwon et al., 2024). By representing a dense weight matrix as the product of two smaller matrices, the computational and memory complexity of linear layers can be substantially reduced. However, traditional low-rank decompositions often exhibit sharp accuracy degradation at high compression ratios (Lee & Kim, 2024), which limits their practicality. To overcome this limitation, structured decompositions such as Monarch (Dao et al., 2022) and BLAST (Lee et al., 2024) have been proposed. They leverage block low-rank (BLR) structures to capture the underlying representation of weight matrices more effectively, thereby better preserving accuracy while offering memory savings and computational reduction.

Despite these algorithmic advances, the expected end-to-end speedups often fail to materialize in practice on GPUs. Performance on modern GPUs is governed by a roofline model (Williams et al., 2009; Yang et al., 2013) that balances memory bandwidth, peak computational throughput, and arithmetic intensity (i.e., the ratio of operations to memory traffic). Figure 1 illustrates

the roofline model of an NVIDIA A40 GPU for 16-bit brain floating-point (BF16) operations. While structured low-rank (LR) decompositions reduce the nominal compute requirements, we first show that they also introduce additional intermediate data movement, particularly in long-sequence scenarios such as the pre-fill stage of large language model (LLM) inference (Jiang et al., 2024; Kaneko & Okazaki, 2023). This shift can move linear layers from the compute-bound regime into the memory-bound regime, creating a gap between algorithmic promise and system-level reality. The issue arises specifically for devices with limited memory subsystems, such as edge GPUs with small L2 caches (4–6 MB) and DRAM based on DDR technology (Jetson Orin Nano) as well as datacenter GPUs (A40). In such cases, (B)LR decompositions paradoxically degrade performance, despite reducing floating-point operations (FLOP) and model size.

In this work, we analyze the performance of LR, Monarch, and BLAST matrix multiplications for efficient transformer-based foundation model inference, with a particular emphasis on long sequences. We identify and characterize key bottlenecks arising from data movement, suboptimal memory layouts, and compiler limitations, and we introduce optimized implementations using Triton (Tillet et al., 2019), an open-source intermediate language designed for writing efficient GPU kernels. Our proposed kernels exploit *partial fusion*, *operation reordering*, and *tailored memory layouts* to mitigate the overheads of (B)LR

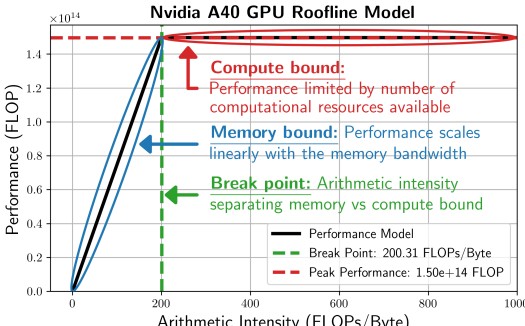

Figure 1: Roofline model of BF16 operations for NVIDIA A40 GPU.

structured matrix multiplications in multi-token inference. Through extensive evaluation, we demonstrate that these optimizations deliver substantial performance gains across diverse transformer models, including GPT2-S, Llama-1/7B, and DiT-XL/2 on both server-grade and edge-class GPUs. Our results establish that BLR-based model compression, when paired with hardware-aware optimizations, offers a viable path toward practical deployment of foundation models in resource-constrained environments. Overall, our contributions can be summarized as follows:

- We provide the first systematic roofline analysis of BLR (Monarch, BLAST) matrix multiplications, showing that while they reduce FLOP, block structures introduce intermediate data movement and uncover PyTorch compiler limitations that push multi-token inference into the memory-bound regime compared to traditional low-rank and dense methods.

- We design Triton kernels with partial fusion, operation reordering, and tensor-core-friendly layouts that eliminate redundant data movement and restore efficiency to BLR inference.

- We release our optimized kernels and benchmark on server and edge-grade GPUs, demonstrating up to $3.76\times$ speedups and $3\times$ model compression over PyTorch CUDA dense baselines with compiler optimizations, fostering reproducibility and rendering structured compression practical.

## 2 BACKGROUND

### 2.1 WEIGHT STRUCTURES

We introduce the weight matrix structures considered in this work, together with their computational properties and modeling capabilities. Let $i$, $o$, $n$, and $r$ denote the number of input features, output features, sequence length, and rank, respectively. The input to all linear layers is $\boldsymbol{X} \in \mathbb{R}^{n \times i}$, and we assume $r \ll i, o$.

**Dense** A dense weight matrix $\boldsymbol{W} \in \mathbb{R}^{i \times o}$ has $i \times o$ parameters, and the corresponding linear layer $\boldsymbol{Y} = \boldsymbol{X}\boldsymbol{W}$ requires $n \times i \times o$ FLOP. Dense matrices can represent arbitrary linear maps and therefore provide the highest expressiveness for foundation models.

**Low-Rank (LR)** Dense weights can be factorized as $\boldsymbol{W} = \boldsymbol{V}\boldsymbol{U}$, where $\boldsymbol{V} \in \mathbb{R}^{i \times r}$ and $\boldsymbol{U} \in \mathbb{R}^{r \times o}$. Instead of materializing $\boldsymbol{W}$, the factorization is stored and used directly in computation. This

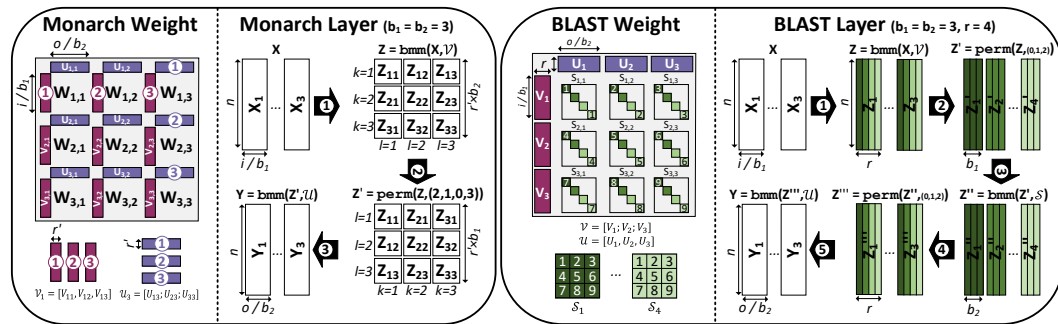

Figure 2: Monarch (left) and BLAST (right) weight parametrization and linear layer execution for $b_1 = b_2 = 3$ blocks and rank $r = 4$.

reduces parameter count to $r(i + o)$ and computation to $nr(i + o)$ FLOP. The low-rank assumption can cause accuracy degradation if the chosen rank $r$ does not capture the true structure of $\boldsymbol{W}$. In practice, $r \ll i, o$ is selected such that $r(i + o) < io$, yielding both memory and computational savings (Wang et al., 2025; Idelbayev & Carreira-Perpiñán, 2020; Kwon et al., 2024).

**Monarch**  Introduced by Dao et al. (2022), Monarch divides a dense weight into $b_2 \times b_1$ blocks, yielding a BLR representation[1] with uniform per-block rank $r'$. Each block $\boldsymbol{W}_{l,k}$ is factorized as

$$\boldsymbol{W}_{l,k} = \boldsymbol{V}_{l,k}\boldsymbol{U}_{l,k} \in \mathbb{R}^{p \times q}, \qquad l \in \{1, \dots, b_1\}, \ k \in \{1, \dots, b_2\}, \ p = i/b_1, \ q = o/b_2.$$

A Monarch layer has $b_1 b_2 r'(p + q)$ parameters. The $k$-th output block $\boldsymbol{Y}_k$ is computed as

$$\boldsymbol{Y}_k = \sum_l \boldsymbol{X}_l \boldsymbol{W}_{l,k}, \qquad \boldsymbol{X}_l \in \mathbb{R}^{n \times p}, \qquad \boldsymbol{Y}_k \in \mathbb{R}^{n \times q},$$

which requires $n b_1 b_2 r'(p + q)$ FLOP in total.

Figure 2 (left) illustrates the execution of a Monarch layer for $b_1 = b_2 = 3$, where a permutation $(b_1 \Leftrightarrow b_2)$ separates two batched matrix multiplications (bmm). In practice, the common setting $b_1 = b_2 = b$ with $r = r'b$ recovers the same complexity as low-rank layers, $r(i + o)$ parameters and $nr(i + o)$ FLOP. Monarch weights are stored as tensors : $\mathcal{V} \in \mathbb{R}^{b_1 \times (r'b_2) \times p}$ and $\mathcal{U} \in \mathbb{R}^{b_2 \times q \times (b_1 r')}$.

**BLAST**  Introduced by Lee et al. (2024), BLAST represents a weight matrix using a generalized BLR structure. Unlike Monarch, each block $\boldsymbol{W}_{l,k}$ shares a pair of matrices $\boldsymbol{V}_l$ and $\boldsymbol{U}_k$ while retaining a unique diagonal matrix $\boldsymbol{S}_{l,k}$ such that the block factorization becomes

$$\boldsymbol{W}_{l,k} = \boldsymbol{V}_l \boldsymbol{S}_{l,k} \boldsymbol{U}_k, \qquad \boldsymbol{V}_l \in \mathbb{R}^{p \times r}, \ \boldsymbol{S}_{l,k} \in \mathbb{R}^{r \times r}, \ \boldsymbol{U}_k \in \mathbb{R}^{r \times q}.$$

This structure generalizes multiple families of structured low-rank matrices. In particular, low-rank and Monarch layers can be recovered by setting the values of $\boldsymbol{S}_{l,k}$ appropriately for all $l$ and $k$.

A BLAST layer has $r(p + q + b_1 b_2)$ parameters. The $k$-th output block $\boldsymbol{Y}_k$ is computed as

$$\boldsymbol{Y}_k = \left(\sum_l (\boldsymbol{X}_l \boldsymbol{V}_l) \boldsymbol{S}_{l,k}\right) \boldsymbol{U}_k, \qquad \boldsymbol{X}_l \in \mathbb{R}^{n \times p}, \qquad \boldsymbol{Y}_k \in \mathbb{R}^{n \times q},$$

which requires $nr(p + q + b_1 b_2)$ FLOP in total.

Typically, $b_1 = b_2 = b \leq 16$, which yields $r(i + o + b^2)$ parameters and $nr(i + o + b^2)$ FLOP. Since $b \ll i, o$, BLAST achieves the same asymptotic savings as low-rank and Monarch layers using a slightly higher compression ratio. Figure 2 (right) illustrates the execution of a BLAST layer for $b_1 = b_2 = 3$ and $r = 4$. In practice, BLAST parameters are stored as $\mathcal{V} \in \mathbb{R}^{b_1 \times p \times r}$, $\mathcal{S} \in \mathbb{R}^{b_1 \times b_2 \times r}$, and $\mathcal{U} \in \mathbb{R}^{b_2 \times r \times q}$.

---

[1]Dao et al. (2022) introduces a "transposed" permutation at the output which we omit here for simplicity.

| Method | Small | | Medium | | | | | Large | |
|---|---|---|---|---|---|---|---|---|---|
| | ViT-B (CF = 3×) | GPT2-S (CF = 1.85×) | DiT-XL/2 (CF = 2×) | | | Llama-3.2-1B (CF = 2×) | | Llama-7B (CF = 2×) | |
| | ImageNet Accuracy (%) | WikiText-103 Perplexity (↓) | FID (↓) | sFID (↓) | IS (↑) | WikiText-2 Perplexity (↓) | Avg. 0-shot Accuracy (%) | WikiText-2 Perplexity (↓) | Avg. 0-shot Accuracy (%) |
| Dense | 78.7 | 20.2 | 9.62 | 6.85 | 121.50 | 11.57 | 56.54 | 9.37 | 66.07 |
| BLAST | 79.3 | 20.7 | 10.45 | 6.72 | 111.05 | 20.10 | 46.37 | 14.21 | 56.23 |
| Monarch | 79.2 | 21.1 | - | - | - | 22.17 | 44.35 | 19.54 | 49.78 |
| Low-Rank | 78.9 | 21.7 | 48.07 | 11.44 | 26.09 | 21.92 | 44.71 | 26.33 | 48.40 |

Table 1: Accuracy of foundation models using different (B)LR model compression factors (CF).

**Accuracy** Lee et al. (2024) evaluate the impact of replacing dense linear layers in foundation models with low-rank, Monarch, and BLAST layers. Results are reported on language and vision tasks using Llama-7/1B (Touvron et al., 2023), GPT2-S (Radford et al., 2019), ViT-B (Dosovitskiy et al., 2021), and DiT-XL/2 (Peebles & Xie, 2022). Language models are evaluated with WikiText-103/2 perplexity and zero-shot classification accuracy on common sense reasoning benchmarks Bisk et al. (2020); Zellers et al. (2019); Sakaguchi et al. (2021); Clark et al. (2019); Mihaylov et al. (2018); Clark et al. (2018). Vision models are evaluated on ImageNet Deng et al. (2009) classification accuracy. Diffusion models are evaluated by generating images with a DDPM sampler (Ho et al., 2020) and computing FID, sFID, and IS against 50,000 ImageNet validation images (step size 250) to quantify generation quality. Table 1 summarizes the results where Monarch mostly improves upon low-rank, while BLAST achieves the best accuracy at the same compression factor (CF).

## 2.2 GPU Performance

**Hardware Architecture** GPUs integrate many CUDA cores for general-purpose parallelism and tensor cores specialized for matrix operations. The memory hierarchy spans high-capacity but high-latency off-chip DRAM and smaller on-chip caches (L1/L2) (Jia et al., 2018). On resource-constrained devices such as the Jetson Orin Nano, the L2 cache is only a few MB (NVIDIA, 2024), so the large activations of foundation models often spill to DRAM between kernels rather than being reused from cache. Even some data center GPUs like the A40 provide just a 6 MB shared L2 cache (NVIDIA, 2022). While L2 is not directly programmable, developers can leverage L1 and shared memory to improve locality and hide latency in software (Choo et al., 2014).

**Execution Model and Programming** Computation is done by many parallel threads organized into thread blocks and scheduled across streaming multiprocessors (SMs). Threads access a hierarchical memory system: high-latency global memory (DRAM), lower-latency but non-programmable L2 cache, explicitly managed shared memory (L1/SRAM) per thread block, and per-thread registers. A typical kernel loads data from global memory into registers or shared memory, performs computations, and writes results back. Kernels are usually written in CUDA, offering fine-grained control but requiring detailed knowledge of GPU architecture to achieve high performance (Che et al., 2008). For example, uncoalesced accesses reduce bandwidth efficiency, making coalesced loads a key optimization (Ryoo et al., 2008). While NVIDIA's proprietary libraries often outperform custom CUDA kernels, recent alternatives like OpenAI's Triton (Tillet et al., 2019) allow developers to write efficient GPU kernels in a Python-like syntax, with the compiler handling optimizations such as shared memory management, warp scheduling, and memory coalescing (Zhou et al., 2025; Li et al., 2025).

**Performance Characteristics** GPU kernels are broadly categorized as compute-bound or memory-bound depending on their arithmetic intensity, $\alpha$ (operations per byte of memory accessed). The roofline model in Figure 1 captures this tradeoff, with the breakpoint $\tilde{\alpha}$ distinguishing compute-bound workloads ($\alpha \geq \tilde{\alpha}$) from memory-bound ones ($\alpha < \tilde{\alpha}$).

## 3 Profiling and Characterizing Bottlenecks

We first conduct a case study on Llama-7B layers to examine how GPUs with limited L2 caches and memory bandwidths exhibit bottlenecks when executing inference with BLR approaches. Using empirical measurements and roofline modeling, we identify when and why these bottlenecks occur, providing insights that motivate our proposal for memory-efficient kernels. We present separate discussions for single-token and multi-token inference.

### 3.1 SINGLE-TOKEN INFERENCE

During single-token inference, typical in the decoding stage of LLMs where $n = 1$, the problem becomes memory-bound (Yuan et al., 2024a). In this setting, memory traffic is dominated by weight movement rather than activations. As a result, compressing weights (e.g., by $2\times$) can nearly double throughput. This trend is evident in Llama-7B layers on the A40 GPU (Figure 3, left). Here, (B)LR methods such as BLAST, Monarch, and traditional low-rank all achieve similar performance compared to dense since the bottleneck lies in weight data movement rather than compute.

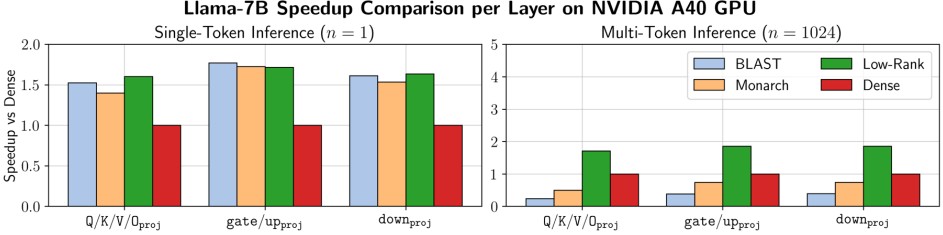

Figure 3: Performance of Llama-7B layers with low-rank methods versus dense, for single-token (left) and multi-token (right) inference on an NVIDIA A40 GPU.

### 3.2 MULTI-TOKEN INFERENCE

Unlike the single-token case, multi-token inference operates on larger input matrices where activation traffic grows with sequence length. This shift exposes a key weakness of (B)LR approaches: all of them generate intermediate outputs absent in the dense baseline. For traditional low-rank, the intermediate is an $n \times r$ matrix; for Monarch, it expands to $b \times n \times r$ with $b$ as large as 16 in Llama-7B; and for BLAST, two such intermediates appear ($b_1 = b_2 = b$). Each of these tensors adds data movement, eroding the theoretical memory and compute savings. Blocked methods are hit especially hard, since the block dimension implies a `bmm`, and both Monarch and BLAST further require permutations on the innermost (contiguous) dimension, creating uncoalesced accesses and throttling memory bandwidth. Figure 3 (right) shows the resulting degradation. Traditional low-rank runs at $0.53$–$0.59\times$ the runtime of dense, roughly consistent with its $2\times$ compression. Monarch slows down, taking $1.14$–$1.68\times$ longer than dense, while BLAST takes $2.63$–$4.31\times$ longer.

| Method | FLOP | Memory (bytes) |
|---|---|---|
| Dense ($D$) | $nio$ | $2 \times (ni + io + no)$ |
| Low-Rank ($LR$) | $nr(i + o)$ | $2 \times (ni + ir + ro + no + \underline{2nr})$ |
| Monarch ($M$) | $nr(i + o)$ | $2 \times (ni + ir + ro + no + \underline{4bnr})$ |
| BLAST ($B$) | $nr(i + o + b^2)$ | $2 \times (ni + ir + ro + rb^2 + no + \underline{8bnr})$ |

Table 2: FLOP and memory traffic for linear layers under different BF16 weight structures.

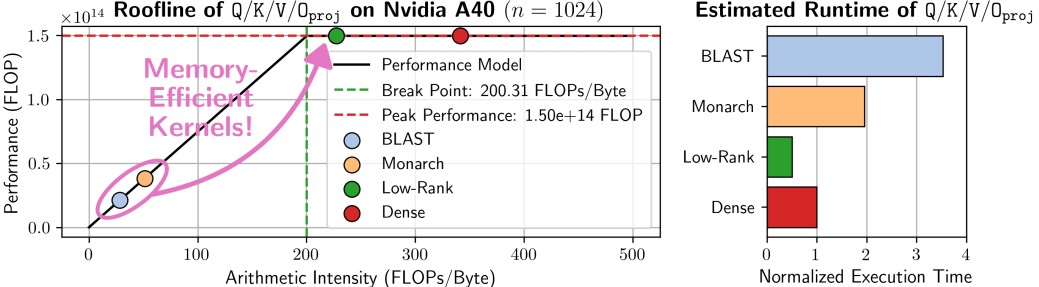

Figure 4: Roofline and runtime estimation of $Q/K/V/O_{\text{proj}}$ during multi-token inference on A40.

To interpret these results, we consider the FLOP and memory traffic for each method summarized in Table 2. The arithmetic intensity $\alpha = \text{FLOP/Memory}$ can then be computed directly from these values for a $Q/K/V/O_{\text{proj}}$ layer during multi-token inference ($b = 16$, $n = 1024$, $r = 1024$, $i = o = 4096$). Figure 4 overlays the resulting arithmetic intensity values on the roofline model and compares

```
1   parfor b₁ in range (0, b₁):
2     parfor n in range(0, n, tₙ):
3       parfor r in range(0, r, tᵣ):
4         # Each thread block executes below
5         b₂ = r // r' (★)
6         r' = r % r' + b₁ * r' (★)
7         acc = zeros((tₙ, tᵣ))
8         for p in range(0, i/b₁, tₚ):
9           x = X[b₁, n : n + tₙ, p : p + tₚ]
10          v = V[b₁, p : p + tₚ, r : r + tᵣ]
11          acc += dot(x, v)
12        Z'[b₂, n : n + tₙ, r' : r' + tᵣ] = acc (★)
```

Figure 5: Pseudo-code for fused permutation and `bmm` Monarch kernel (❷).

estimated runtimes. Dense ($\alpha_D$) and traditional low-rank ($\alpha_{LR}$) lie above the roofline breakpoint and are compute-bound, while Monarch ($\alpha_M$) and BLAST ($\alpha_B$) fall below it and are memory-bound, mirroring the empirical results. Therefore, multi-token inference exposes a fundamental limitation: BLR methods are undermined by intermediate data movement and poor memory access patterns. To mitigate these issues, we propose *fused* and *memory-efficient* kernels in Triton that avoid redundant memory trips and reorganize computation to exploit better layouts of intermediate tensors.

# 4 MEMORY-EFFICIENT KERNELS

**Full Fusion**  Kernel fusion integrates consecutive kernels into one. For Monarch and BLAST, we first explored fully fusing the `bmm` kernels, building on prior work for low-rank layers (Sun et al., 2024; Al Awar et al., 2025). In Triton, matrix multiplications are parallelized via 2-D output tiling, where each thread block iterates over the inner dimension and loads operand tiles into shared memory for tensor core computation via the `dot()` operator. In the context of full fusion, this tiling leads to redundant weight loads and recomputation of intermediates, often making fusion slower than launching separate kernels. Using 1-D tiles avoids redundancy but restricts rank and parallelism, yielding speedups only for very small ranks (e.g., $\leq 128$) that correspond to extreme compression ratios (e.g., $\geq 8\times$ for `Q/K/V/O_proj` in Llama-7B). An evaluation of traditional low-rank is provided in Appendix A.1, which shows that larger ranks are slower than dense or fail due to shared memory limits, while small-rank gains quickly diminish with output size due to limited parallelism. Given these limitations, we turn to *partial fusion*, where only permutations or subsets of `bmm` are fused to reduce memory traffic. We next outline kernel-specific optimizations for Monarch and BLAST, whose different parameterizations motivate distinct strategies.

**Monarch**  Optimizations ❶, ❷, and ❸ described below are intended to be employed together to provide additive efficiency benefits to Monarch linear layers during inference.

❶ *Re-layout of $\mathcal{V}$.* In the original code, $\mathcal{V}$ is stored as a $(b_1, r'b_2, p)$ tensor with the middle dimension contiguous along $b_2$ then $r'$. Meanwhile, $\mathcal{U}$ is stored as a $(b_2, q, b_1r')$ tensor with the innermost dimension contiguous along $r'$ then $b_1$. Multiplying the input batches with $\mathcal{V}$ after transposing its last two dimensions produces a $(b_1, n, r'b_2)$ tensor, after which two permutations become necessary: $r' \leftrightarrow b_2$ first, then $b_2 \leftrightarrow b_1$. In practice, this results in two separate kernel launches, each cloning the tensor into a different layout and incurring uncoalesced loads because it targets the innermost (contiguous) dimension. The first optimization is therefore to modify the memory layout of $\mathcal{V}$ so the middle dimension is contiguous along $r'$ first, then $b_2$. Since $\mathcal{V}$ is a static weight, this re-layout can be performed once before inference, eliminating the unnecessary $r' \leftrightarrow b_2$ permutation.

❷ *Permutation fusion.* After optimally re-laying out $\mathcal{V}$, we fuse the $b_2 \leftrightarrow b_1$ permutation with the first `bmm` in a single Triton kernel. The fused kernel computes $b_1 \times t_n \times t_r$ output tiles, where $t_n$ and $t_r$ are the tile sizes along $n$ and $r = r'b_2$, respectively. The permutation is implemented by first calculating the index $b_2$ (★, see Figure 5), then adjusting the innermost index $r'$ by the offset of the corresponding block indexed at $b_1$ (★), and writing out the output using the swapped indices (★). These three steps are highlighted in the pseudo-code shown in Figure 5.

❸ *Avoiding the final permutation.* After computing the Monarch linear layer, a final permutation is applied to the output, transforming its shape from $(b_2, n, q)$ to $(n, q, b_2)$. Unlike the simpler stride change to $(n, b_2, q)$, this transformation requires a full kernel launch. The additional permutation is

```
1  parfor n in range(0, n, t_n):
2    parfor r in range(0, r, t_r):
3      # Each thread block executes below
4      z'' = zeros((b_2, t_n, t_r))
5      for b_1 in range(0, b_1): (★)
6        s = expand_dims(S[b_1, :, r : r + t_r], 1) (★)
7        z' = zeros((t_n, t_r))
8        for p in range(0, i/b_1, t_p):
9          x = X[b_1, n : n + t_n, p : p + t_p]
10         v = V[b_1, p : p + t_p, r : r + t_r]
11         z' += dot(x, v)
12        z' = expand_dims(z', 0) (★)
13        z'' += s * z' (★)
14      Z''[:, n : n + t_n, r : r + t_r] = z'' (★)
```

Figure 6: Pseudo-code for BLAST partial fusion (❹).

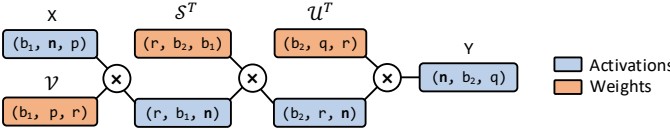

Figure 7: Compute diagram for BLAST permutation-only fusion (❺).

unavoidable if the output of the Monarch linear layer is consumed by a residual connection or split into multiple heads. However, if the output is immediately multiplied by a static weight, we can pre-permute the rows of that weight offline and avoid running this kernel at inference time.

**BLAST** Optimizations ❹ and ❺ are applied separately as they represent distinct strategies to improve the efficiency of BLAST linear layers during inference.

❹ *Partial fusion of* `bmm`. This optimization eliminates both the intermediate permutation between $\mathcal{V}$ and $\mathcal{S}$ and the materialization of the first `bmm` output in global memory. Instead of assigning each thread block to a separate $b_1$ batch, we loop over the $b_1$ dimension within each thread block to compute an output tile of $Z$ (★, see Figure 6). This restructuring is required because the second `bmm` reduces along $b_1$. If $b_1$ were distributed across multiple thread blocks, the threads would not be able to share the data needed for the reduction. Within each $b_1$ loop iteration, we load a $(b_2, t_r)$ tile of $\mathcal{S}$ stored here as a $(b_1, b_2, r)$ tensor. We reshape it to $(b_2, 1, t_r)$ and broadcast it with the $(1, t_n, t_r)$ output from the first `bmm` (★). This allows the second `bmm` to be expressed as an accumulated batched outer product across $t_r$ (★). The results are accumulated in $Z''$ with shape $(b_2, t_n, t_r)$, avoiding the large intermediate tensor required in the baseline (★). The trade-off however is that tensor cores cannot be used for the second `bmm`. The overall procedure is highlighted in Figure 6.

❺ *Permutation-only fusion with tensor core optimization.* The earlier strategy (❹) mapped the second `bmm` to CUDA cores rather than tensor cores, sacrificing up to $16\times$ higher throughput (Dao, 2023). This tradeoff could negate the FLOP savings from BLAST. To preserve tensor-core execution, one alternative is to eliminate only the costly permutations, but this is challenging because BLAST swaps the outer and innermost dimensions. Directly writing into the target layout of the next `bmm` leads to uncoalesced stores, as the batch dimension split across thread blocks cannot share data. Our key insight is that transposing the `dot()` output before storing is inexpensive in Triton. We therefore reorder the computation as shown in Figure 7. Instead of right-multiplying by $\mathcal{S}$ and $\mathcal{U}$, we transpose their first and last dimensions to obtain $\mathcal{S}^T$ and $\mathcal{U}^T$, multiply from the left, and transpose intermediate output tiles within each kernel. This keeps $n$ contiguous, while $r$, $b_1$, and $b_2$ are successively exposed as the batch dimensions across three kernels, each implementing a transposed `bmm` with outer-dimension reordering. Reordering eliminates permutation overhead and maintains high tensor-core utilization via Triton's `dot()`, a level of efficiency that, to our knowledge, neither `einsum` nor PyTorch compiler-guided methods can match.

## 5 Experiments and Results

**Evaluation Setup** We evaluate our memory-efficient kernels against baseline implementations from the BLAST (Lee et al., 2024) and Monarch (Dao et al., 2022) repositories. Prior work focused

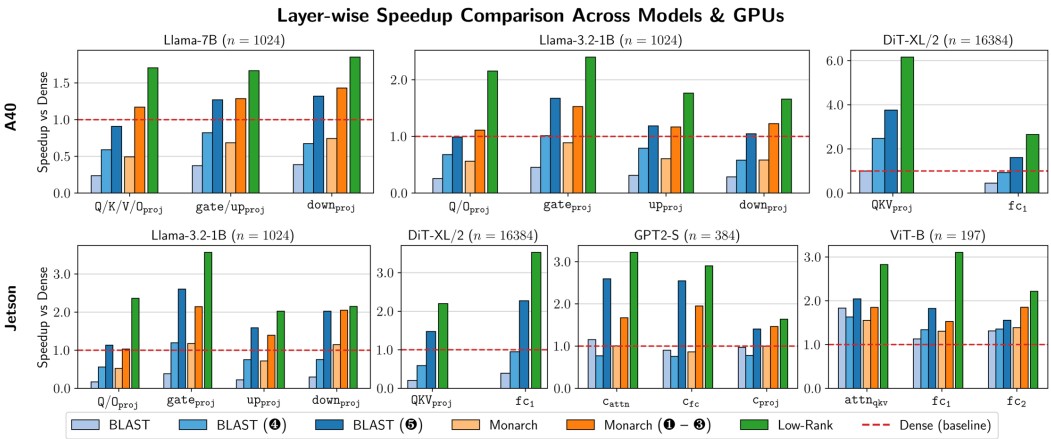

Figure 8: Layer-wise performance comparison across BLR methods and GPUs.

on accuracy across a variety of language and vision domain tasks. As shown in Table 1, low-rank performs worst, Monarch improves upon low-rank, and BLAST achieves the highest accuracy with the same model compression factor (CF, ranging from $1.8$ to $3\times$). Our evaluation complements these results with detailed performance benchmarking on the same set of models while additionally including Llama-3.2-1B for broader coverage. Details regarding Llama-3.2-1B training are provided in Appendix A.3 with accuracies reported in Table 1.

We conduct experiments on two hardware platforms. For mid to large-scale models (Llama-7B, DiT-XL/2, Llama-3.2-1B), we use an NVIDIA A40 with 40GB of memory. For mid to small-scale models (Llama-3.2-1B, DiT-XL/2, GPT2-S, ViT-B), we evaluate on the Jetson Orin Nano with 8GB of memory. Note that Llama-7B does not fit on the Jetson device. All experiments use batch size of 1, with $n$ determined by the model and application. Models are evaluated in BF16, and results are validated against original PyTorch implementations. Baselines leverage both Triton's auto-tuner and `torch.compile()` for fair comparison. For language models (GPT2-S, Llama), we report prefill throughput, for diffusion models, we benchmark inference at a single step, and for vision models, we measure standard forward inference. Experimental details are provided in Appendix A.3.

**Layer-wise Breakdown** Figure 8 summarizes layer-wise speedups. Our optimized BLAST kernel (❺) consistently outperforms both the BLAST and Monarch baselines across all architectures. Notably, ❺ delivers up to $7.15\times$ speedup over its baseline for the $\text{QKV}_{\text{proj}}$ layer of DiT-XL/2 on Jetson, and up to $2.95\times$ over Monarch for the $\text{c}_{\text{fc}}$ layer of GPT2-S on Jetson. Our optimized Monarch kernel (❶ – ❸) also provides meaningful gains, achieving $1.46$–$2.37\times$ speedups across layers relative to its baseline. Since BLAST also achieves higher accuracy than Monarch overall, ❺ represents the best balance of accuracy and efficiency. Most importantly, ❺ outperforms dense by $1.13$-$3.76\times$ in $> 90\%$ of cases where its baseline falls short, proving competitive with highly tuned dense kernels as well as other BLR baselines. We highlight an additional observation. BLAST kernels employing optimization ❹ are consistently worse than those employing ❺, and in some cases worse than baseline BLAST, such as GPT2-S on Jetson, because the second `bmm` in ❹ is mapped as batched outer product running on CUDA cores while ❺ leverages tensor cores for this `bmm`.

**End-to-End Comparison** Figure 9 reports end-to-end inference results with dense linear layers replaced by BLAST, Monarch, or low-rank layers, using either baseline or optimized BLR implementations. Details regarding the layers replaced in each architecture are provided in Appendix A.2. To minimize CPU overhead and improve scheduling efficiency, we apply `torch.compile()` to the entire network in each case, enabling CUDA graph execution. Since BLR linear layers account for only part of the total runtime, overall speedups are naturally smaller than layer-wise gains, especially in long-sequence workloads such as DiT-XL/2 with 16K tokens where attention dominates.

Nevertheless, BLAST-based models using ❺ achieve substantial acceleration over models that use its baseline and the Monarch baseline. Relative to the BLAST baseline, ❺ provides $3.05\times$ speedup on Llama-7B/A40, $2.48\times$ on Llama-3.2-1B/A40, $3.68\times$ on Llama-3.2-1B/Jetson, and $1.36\times$ on

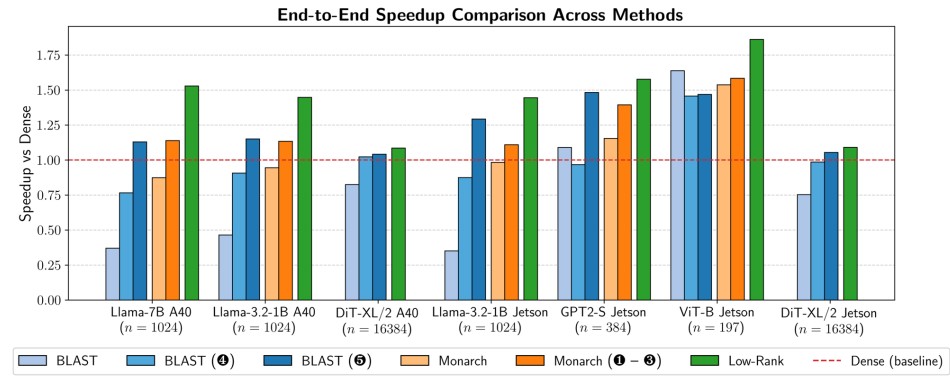

Figure 9: End-to-end inference performance across models and platforms.

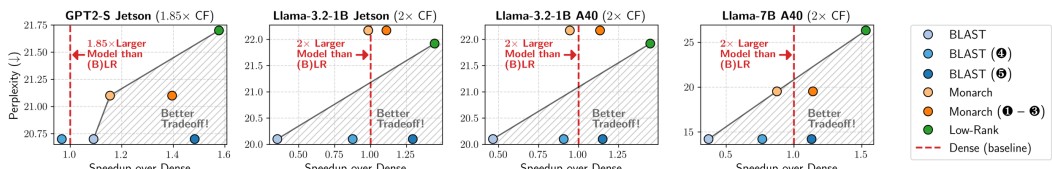

Figure 10: Tradeoff between perplexity and speedup over dense compared across methods.

GPT2-S/Jetson end-to-end. Even in settings where attention dominates like DiT-XL/2 (Yuan et al., 2024b), ❺ reduces overall inference time by $1.4\times$ on the Jetson compared to the BLAST baseline. Some cases such as ViT-B show a different trend where most of the gain comes from applying `torch.compile()` to the entire model, which already yields speedups over the dense baseline comparable to those of low-rank. In this setting, ❺ does not provide additional benefit and can even regress performance, while ❶ – ❸ offer little improvement. The configuration itself (with only $b=3$ blocks for BLAST, $b=4$ for Monarch, and rank 128) is small and particularly well-suited for `torch.compile()` to produce an optimized baseline.

Finally, ❺ not only outperforms dense across all tested models by up to $1.48\times$ but also approaches the speed of low-rank while maintaining higher accuracy. This establishes it as the most effective option when, for instance, end-to-end gains from ❶ – ❸ are comparable. This is made evident in Figure 10 where both ❺ and ❶ – ❸ provide a better tradeoff than low-rank and BLR baselines for language models between perplexity and speedup over dense.

## 6 CONCLUSION

This work shows that while prior studies of BLR foundation models focused on modeling accuracy and single-token inference performance, their speedup benefits often vanish in multi-token settings, especially on resource-constrained GPUs. Through a detailed roofline analysis and memory-efficient Triton kernels, we bridge the gap between reduced FLOP and realized speedups using techniques such as partial fusion, operation re-ordering, and optimized memory layouts. This in turn enables practical deployment of BLR-compressed foundation models at a level that, to our knowledge, current PyTorch compiler-guided implementations cannot achieve.

**Limitation and Future Work** Our optimized BLAST and Monarch routines still lag behind low-rank decompositions in terms of speed, a limitation rooted in their blocked structure, which generates more intermediate outputs. This overhead, however, could be mitigated by future techniques such as intermediate activation quantization, provided accuracy is preserved and target devices support mixed-precision tensor core operations. Recent works are actively exploring activation quantization (Ashkboos et al., 2024; Liu et al., 2025; 2024), either through co-design during training or post-training calibration, and integrating such methods with BLR could further reduce overheads. Finally, while our experiments on billion-parameter models were constrained to partial re-training (only 400-4000 training steps) due to limited compute resources, extended re-training, as was feasible for smaller models ($> 10^6$ training steps), could further narrow the accuracy gap to dense baselines.

# 7 REPRODUCIBILITY STATEMENT

To ensure reproducibility of our results, the authors have undertaken the following measures:

**Code Availability** We provide an anonymized repository containing all code necessary to reproduce our experimental results in the supplementary material. Upon acceptance, we commit to making our complete implementation publicly available, including optimized Triton kernels and benchmarking scripts.

**Experimental Details** We provide comprehensive details regarding our evaluation setup, including hardware specifications, software versions, model architectures, and hyperparameters in Section 5 and Appendix A.3. Layer-specific details for each model are documented in Appendix A.2.

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

## A APPENDIX

### A.1 FULL FUSION

As discussed in the main text, fully fusing matrix multiplication kernels suffers from fundamental limitations due to the way matrix multiplications are parallelized with 2-D output tiling. In the low-rank setting with matrices $V$ and $U$, two main issues arise. First, thread blocks computing neighboring output tiles along the same rows redundantly load the entire $V$ matrix. Second, they also redundantly compute tiles of the intermediate product $XV$, which undermines the benefit of low-rank factorization since its purpose is to reduce FLOP. Switching to 1-D output tiling eliminates redundant computation, but this comes at the cost of restricting both the feasible rank values and the degree of parallelism across columns. In practice, full fusion only works for small ranks, as the shared memory budget is quickly exhausted when the rank dimension $r$ is fully loaded as a tile ($t_r = r$), limiting the number of active thread blocks per streaming multiprocessor. Figure 11 illustrates these tradeoffs. Our Triton implementation of fully fused low-rank highlights the strong dependence of performance on $r$: for $r = 256$ (Figure 11, right), full fusion is consistently slower than dense across all feature dimensions, while for $r = 128$ (Figure 11, left), speedups appear but only for small output dimensions. As output dimension grows, the baseline low-rank implementation increasingly benefits from parallelism across the second dimension, which is sacrificed under 1-D tiling. Consequently, fully fused low-rank underperforms in these regimes.

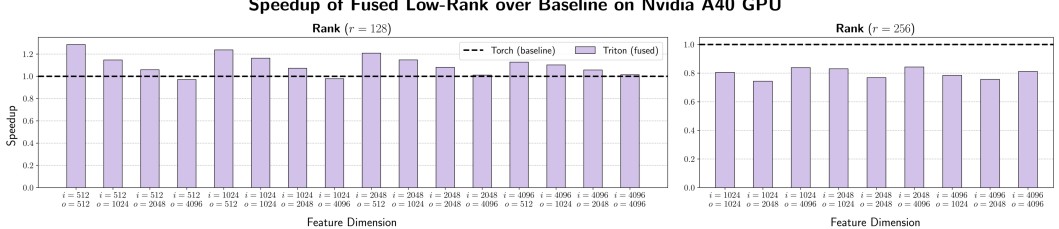

Figure 11: Speedup of Triton fused low-rank matrix multiplication over the PyTorch low-rank baseline on NVIDIA A40 GPU. Results are shown across different input/output feature dimensions for two fixed ranks: $r = 128$ (left) and $r = 256$ (right).

### A.2 LAYER DETAILS

In this section, we provide detailed configurations for all layers that were replaced with (B)LR counterparts in the evaluated models. This includes the rank, number of blocks, input/output feature dimensions, and the number of occurrences of each layer type within the network. A summary of

these details is presented in Table 3. Note that for DiT-XL/2, $\text{adaLN}_{\text{proj}}$ was replaced with a (B)LR counterpart to compress the model, but it was not included in the layer-wise benchmarking results reported in Section 5, as it processes a single token rather than the 16K tokens used in other layers.

| Model | Layer | Input ($i$) | Output ($o$) | Indices | Method | ($r$, $b$) |
|---|---|---|---|---|---|---|
| Llama-7B | $\text{Q/K/V/O}_{\text{proj}}$ | 4096 | 4096 | $0 - 31$ | Low-Rank | $(1024, \, -)$ |
| | | | | | Monarch | $(1024, \, 16)$ |
| | | | | | BLAST | $(1024, \, 16)$ |
| | $\text{gate/up}_{\text{proj}}$ | 4096 | 11008 | $0 - 31$ | Low-Rank | $(1488, \, -)$ |
| | | | | | Monarch | $(1536, \, 16)$ |
| | | | | | BLAST | $(1488, \, 16)$ |
| | $\text{down}_{\text{proj}}$ | 11008 | 4096 | $0 - 31$ | Low-Rank | $(1488, \, -)$ |
| | | | | | Monarch | $(1536, \, 16)$ |
| | | | | | BLAST | $(1488, \, 16)$ |
| Llama-3.2-1B | $\text{Q/O}_{\text{proj}}$ | 2048 | 2048 | $0 - 31$ | Low-Rank | $(256, \, -)$ |
| | | | | | Monarch | $(256, \, 16)$ |
| | | | | | BLAST | $(256, \, 16)$ |
| | $\text{gate}_{\text{proj}}$ | 2048 | 8192 | $0 - 31$ | Low-Rank | $(512, \, -)$ |
| | | | | | Monarch | $(512, \, 16)$ |
| | | | | | BLAST | $(512, \, 16)$ |
| | $\text{up}_{\text{proj}}$ | 2048 | 8192 | $0 - 31$ | Low-Rank | $(768, \, -)$ |
| | | | | | Monarch | $(768, \, 16)$ |
| | | | | | BLAST | $(768, \, 16)$ |
| | $\text{down}_{\text{proj}}$ | 8192 | 2048 | $0 - 31$ | Low-Rank | $(768, \, -)$ |
| | | | | | Monarch | $(768, \, 16)$ |
| | | | | | BLAST | $(768, \, 16)$ |
| GPT2-S | $\text{c}_{\text{attn}}$ | 768 | 2304 | $0 - 11$ | Low-Rank | $(192, \, -)$ |
| | | | | | Monarch | $(192, \, 4)$ |
| | | | | | BLAST | $(192, \, 6)$ |
| | $\text{c}_{\text{fc}}$ | 768 | 3072 | $0 - 11$ | Low-Rank | $(192, \, -)$ |
| | | | | | Monarch | $(192, \, 4)$ |
| | | | | | BLAST | $(192, \, 6)$ |
| | $\text{c}_{\text{proj}}$ | 3072 | 768 | $0 - 11$ | Low-Rank | $(192, \, -)$ |
| | | | | | Monarch | $(192, \, 4)$ |
| | | | | | BLAST | $(192, \, 6)$ |
| ViT-B | $\text{attn}_{\text{qkv}}$ | 768 | 2304 | $0 - 11$ | Low-Rank | $(128, \, -)$ |
| | | | | | Monarch | $(128, \, 4)$ |
| | | | | | BLAST | $(128, \, 3)$ |
| | $\text{fc}_1$ | 768 | 3072 | $0 - 11$ | Low-Rank | $(128, \, -)$ |
| | | | | | Monarch | $(128, \, 4)$ |
| | | | | | BLAST | $(128, \, 3)$ |
| | $\text{fc}_2$ | 3072 | 768 | $0 - 11$ | Low-Rank | $(128, \, -)$ |
| | | | | | Monarch | $(128, \, 4)$ |
| | | | | | BLAST | $(128, \, 3)$ |
| DiT-XL/2 | $\text{QKV}_{\text{proj}}$ | 1152 | 3456 | $0 - 27$ | Low-Rank | $(384, \, -)$ |
| | | | | | BLAST | $(384, \, 9)$ |
| | $\text{fc}_1$ | 1152 | 4608 | $0 - 27$ | Low-Rank | $(256, \, -)$ |
| | | | | | BLAST | $(256, \, 9)$ |
| | $\text{adaLN}_{\text{proj}}$ | 1152 | 6912 | $0 - 27$ | Low-Rank | $(256, \, -)$ |
| | | | | | BLAST | $(256, \, 9)$ |

Table 3: Layer configurations for dense layers replaced by low-rank, Monarch, and BLAST counterparts across evaluated models.

## A.3 EXPERIMENTAL DETAILS

**Benchmarking** We conducted our benchmarking experiments using Python 3.12.8, PyTorch 2.8.0, Triton 3.4.0, and CUDA 12.6.3 on the NVIDIA A40 GPU. For the Jetson Orin Nano 8GB, we used JetPack 6.2 with L4T 36.4.3, CUDA 12.6.11, PyTorch 2.6.0, and Triton 3.2.0. Latency of individual layers was measured with Triton's `do_bench()` utility, which executes the targeted layer multiple times under controlled conditions and reports averaged runtime. End-to-end model inference latency was measured using PyTorch's benchmarking utilities. To eliminate

cold-start effects such as kernel compilation and cache population, we first performed several warm-up passes. During timing, inference ran under `torch.no_grad()` to disable gradient tracking, and we invoked `torch.cuda.synchronize()` to account for asynchronous CUDA execution. Measurements were collected with `torch.utils.benchmark.Timer()`, which repeatedly executes the forward pass for a specified number of iterations. As discussed in Section 5, the model was compiled with `torch.compile()`, so the reported results reflect execution under CUDA graph capture with reduced CPU dispatch overhead.

**Kernel Autotuning**  As illustrated by the pseudo-code in Figures 6 and 7, GPU kernels define tile sizes that partition the problem into parallelizable chunks. Tile sizes are typically chosen as powers of two (commonly between 32 and 256) to align with hardware constraints. Beyond tile size, other hyperparameters such as the number of threads per block and the number of pipelining stages significantly affect kernel performance, memory requirements, and scheduling. Triton provides an `autotune` decorator that explores candidate configurations for these hyperparameters. The auto-tuner sweeps through different values, executes each configuration once to evaluate performance, and caches the best-performing settings for subsequent runs, conditioned on a sensitivity list of parameters. If any of these parameters change, the autotuner is re-run. The hyperparameter values swept for each kernel are documented in the source code.

**Llama-3.2-1B Compression and Re-training**  We employed the Llama-3.2 (Grattafiori et al., 2024) model with 1.24B parameters as one of the representative medium-sized models. The model was compressed by 50% using low-rank, Monarch, and BLAST weight parameterizations. Specifically, we replaced the query and output projection weights in the attention modules, as well as the up projection, gate projection, and down projection weights in the feed-forward network modules.

The rank and number of blocks used in each experiment are reported in Table 3. For low-rank compression, the weights were factorized via singular value decomposition (SVD). For Monarch compression, we applied block-wise SVD, where each block had rank $r' = \frac{r}{b}$. Following (Lee et al., 2024), BLAST compression was obtained by applying 300 steps of preconditioned gradient descent to factorize the weights into BLAST factors. All three methods reduced the model size to 0.6B parameters.

As in (Lee et al., 2024), the compressed models were re-trained. Specifically, the compressed weights were fine-tuned for 4000 steps on a subset[2] of the SlimPajama dataset (Soboleva et al., 2023), using a learning rate of $8 \times 10^{-4}$ with linear decay scheduling.

A.4  USE OF LARGE LANGUAGE MODELS

LLMs were used to aid in wording and polishing the writing. All substantive ideas, experiments, and analyses are the authors' own.

---

[2]`https://huggingface.co/datasets/DKYoon/SlimPajama-6B`

