# OpenReview forum: "Memory-Efficient Acceleration of Block Low-Rank Foundation Models on Resource Constrained GPUs"
_ICLR.cc/2026/Conference — ICLR 2026 Conference Desk Rejected Submission_

### Official Review · Reviewer_Ak8A · 2025-10-28

**Soundness:** 3
**Presentation:** 3
**Contribution:** 3
**Rating:** 6
**Confidence:** 3

**Summary:**

The paper uses roofline analysis to show that block low-rank (BLR) methods achieve theoretical savings and optimize single-token inference, and on the other hand multi-level token inference becomes memory bound. To solve this, the paper introduces kernels written in Triton for Monarch and BLAST layers. The paper analyses memory constrained GPUs, Jetson Orin Nano, and A40. Their kernels deliver both speedups and model size compression over dense baselines in PyTorch.

**Strengths:**

The paper is clearly written and well structured, making the technical contributions easy to follow. On quality, the empirical results are strong: the method achieves up to 3.76× speedup over the dense baseline and demonstrates an approximately 3× improvement in model size, indicating meaningful efficiency gains. The implementation choices further support robustness: the kernels are written in Triton, positioning the work to benefit from ongoing compiler and hardware-backend optimizations. On originality, the paper thoughtfully combines roofline modeling with Triton-level kernel design to guide sparsity-aware optimizations, an effective synthesis of performance analysis and low-level implementation practice. Finally, on significance and clarity, the roofline analysis is used not just descriptively but to justify concrete design decisions in the kernels, helping readers understand why the approach works and where it is likely to generalize. Overall, the paper offers a clear, well-motivated, and practically impactful contribution.

**Weaknesses:**

Significance/motivation. The case for block-low-rank (BLR) matrices is under-motivated. It remains unclear how frequently BLR kernels arise in real-world workloads and whether practitioners deploy them at scale.

Evaluation scope. All experiments use batch size = 1, which limits generality. Performance on modern accelerators often changes with batch size, sequence length, block size/rank, and tensor shapes. Results sweeping batch size, sequence length, and BLR configurations would make the findings more robust.

Baselines. Comparisons appear limited to PyTorch implementations. For a fair performance picture, one could include vendor-optimized libraries, e.g., CUTLASS/cuBLASLt, TensorRT where applicable).

Portability/generalization. Results focus on Jetson Orin Nano and A40. It is unclear whether the gains transfer to more common data-center GPUs , such as, A100, H100, B100 NVIDIA GPUs.

**Questions:**

1. How exactly is the speedup evaluated? For example, [1] in Rule 1 suggests: “When  publishing parallel speedup, report if the base case is a single parallel process or best serial execution, as well as the absolute execution performance of the base case. “
Could you report your base case of the dense baseline, and clarify your evaluation according to this suggestion of benchmarking?

2. Your evaluation targets Jetson Orin Nano and A40. What performance should readers expect on A100/H100/B100 or similar widely used NVIDIA GPUs? Are there kernel or compiler assumptions that limit portability (e.g., tensor-core MMA variants, shared-memory pressure, register usage, or Triton autotuning constraints)?

3. Since BLR can be used for training from scratch, do your findings carry over to training workloads? Can the proposed kernels be used for backward passes, and if so, what speedups versus dense baselines should one expect during pre-training or fine-tuning? If not currently supported, what are the blockers?

[1] Hoefler, Torsten, and Roberto Belli. "Scientific benchmarking of parallel computing systems: twelve ways to tell the masses when reporting performance results." Proceedings of the international conference for high performance computing, networking, storage and analysis. 2015.

---

> ### Author Response · Authors · 2025-12-03
>
> We thank the reviewer for the positive feedback and remarks. We address each of the reviewer’s comments and questions in the following.
>
> **Q1:** *Significance/motivation. The case for block-low-rank (BLR) matrices is under-motivated. It remains unclear how frequently BLR kernels arise in real-world workloads and whether practitioners deploy them at scale.*
>
> **A1:** We understand the reviewer’s concern. However, the block low-rank families have strong potential to be universal components not only in deep learning but also in machine learning and signal processing. Improving the memory efficiency of Monarch and BLAST matrix multiplications, as done in this work, will further encourage their adoption.
>
> From the deep learning standpoint, [1] found that transformers with Monarch or Block Tensor-Train (BTT) weight matrices follow better scaling laws than networks with dense weight matrices under the same FLOP budget. BLAST also follows the conditions identified by [1] for outperforming the scaling laws of dense models. This suggests that networks with the block low-rank families (Monarch, BTT, and BLAST) can serve as standard components in deep learning, as they can deliver higher accuracy given a fixed training budget.
>
> From the machine learning (ML) and signal processing (SP) perspective, Monarch and BLAST matrices can provide a GPU-friendly implementation of various transforms prevalent in SP and ML in general, including discrete Fourier transforms (DFT), and discrete sine/cosine transforms (DST/DCT) [2]. Since many ML/SP research and applications rely on GPUs, Monarch and BLAST can be adopted to utilize GPU resources efficiently. For example, [3]  implemented a DFT using Monarch matrices to leverage tensor cores on NVIDIA GPUs for higher throughput.
>
> We also emphasize that this paper substantially improves the memory efficiency of Monarch and BLAST matrix multiplication routines, which will further support their adoption in the ML, deep learning, and signal processing communities.
>
> [1] Qiu et al. “Compute Better Spent: Replacing Dense Layers with Structured Matrices.” ICML24.
>
> [2] Dao et al. “Monarch: Expressive Structured Matrices for Efficient and Accurate Training.” ICML22.
>
> [3] Fu et al. "FlashFFTConv: Efficient Convolutions for Long Sequences with Tensor Cores." ICLR24.
>
> **Q2:** *Evaluation scope. All experiments use batch size = 1, which limits generality. Performance on modern accelerators often changes with batch size, sequence length, block size/rank, and tensor shapes. Results sweeping batch size, sequence length, and BLR configurations would make the findings more robust.*
>
> **A2: (1/2)** We agree with the reviewer that performance on modern accelerators can vary with batch size, sequence length, block size/rank, and tensor shapes. Our work addresses the bottlenecks of block low-rank matrix multiplications when the sequence length is greater than 1. Increasing the batch size would add more rows to the input activation matrix and because the batch and sequence dimensions can be flattened into a single dimension without incurring any data movement, our kernel does indeed support larger effective batch sizes.
>
> The scope of this paper is inference-time evaluation of compressed block low-rank (BLR) foundation models, so we adopt the tensor shapes and BLR configurations used in prior work, where accuracy numbers were established. These settings are summarized in Table 1. While our kernels can support full sweeps over batch size, sequence length, and BLR parameters, such a comprehensive exploration was beyond the paper’s primary goal of analyzing BLR bottlenecks in realistic foundation-model deployments.

---

> ### Author Response · Authors · 2025-12-03
>
> **A2 (2/2):** To further address the reviewer’s concern, we now include additional layer-wise benchmarking for Llama-7B on A40 and Llama-3.2-1B on both A40 and Jetson Orin Nano, sweeping the product of sequence length $n$ and batch size $B$. Across all settings, we observe consistent speedups relative to baseline implementations as the combined dimension $ntimesB$ increases. All times are reported in milliseconds.
>
> **Llama-7B A40 ($n\times B=2048$)**
>
> |Layer|BLAST|**BLAST (4)**|**BLAST (5)**|Monarch|**Monarch (1-3)**|Low-Rank|Dense|
> |---|---|---|---|---|---|---|---|
> |Q/K/V/O_proj|2.66029|1.0028|0.635813|1.25135|0.496798|0.344755|0.641272|
> |gate_up_proj|4.47584|2.02607|1.28826|2.45583|1.31464|0.933683|1.55064|
> |down_proj|4.35128|2.57358|1.25232|2.26833|1.2011|0.859723|1.59662|
>
> **Llama-7B A40 ($n\times B=3072$)**
>
> |Layer|BLAST|**BLAST (4)**|**BLAST (5)**|Monarch|**Monarch (1-3)**|Low-Rank|Dense|
> |---|---|---|---|---|---|---|---|
> |Q/K/V/O_proj|4.04541|1.47877|1.0078|1.83753|0.721371|0.484518|0.880189|
> |gate_up_proj|6.80393|3.05064|2.01042|3.69223|1.97672|1.24603|2.3027|
> |down_proj|6.57237|3.87153|1.84765|3.40627|1.80429|1.21999|2.4194|
>
> **Llama-7B A40 ($n\times B=4096$)**
>
> |Layer|BLAST|**BLAST (4)**|**BLAST (5)**|Monarch|**Monarch (1-3)**|Low-Rank|Dense|
> |---|---|---|---|---|---|---|---|
> |Q/K/V/O_proj|5.26776|1.93972|1.18596|2.44417|0.95868|0.68814|1.18958|
> |gate_up_proj|9.0836|4.08229|2.5559|4.92328|2.66842|1.69923|3.0803|
> |down_proj|8.72602|5.18326|2.48133|4.55836|2.4118|1.63807|3.17774|
>
> **Llama-3.2-1B A40 ($n\times B=2048$)**
>
> |Layer|BLAST|BLAST (4)|**BLAST (5)**|Monarch|**Monarch (1-3)**|Low-Rank|Dense|
> |---|---|---|---|---|---|---|---|
> |Q/O_proj|0.680783|0.257647|0.174224|0.302348|0.162906|0.0669543|0.17341|
> |gate_proj|1.48715|0.629164|0.377084|0.757709|0.439594|0.265954|0.617127|
> |up_proj|2.19009|0.842765|0.541386|1.10737|0.580459|0.373958|0.619943|
> |down_proj|2.09521|0.984984|0.541541|1.00143|0.471404|0.406887|0.639169|
>
> **Llama-3.2-1B A40 ($n\times B=3072$)**
>
> |Layer|BLAST|**BLAST (4)**|**BLAST (5)**|Monarch|**Monarch (1-3)**|Low-Rank|Dense|
> |---|---|---|---|---|---|---|---|
> |Q/O_proj|1.02848|0.346537|0.26855|0.438356|0.251765|0.0990953|0.258886|
> |gate_proj|2.259|0.9217|0.59453|1.12284|0.651137|0.395273|0.887642|
> |up_proj|3.25041|1.22387|0.858221|1.6442|0.855512|0.496531|0.887656|
> |down_proj|3.10077|1.4657|0.831302|1.47277|0.677004|0.491707|0.965398|
>
> **Llama-3.2-1B A40 ($n\times B=4096$)**
>
> |Layer|BLAST|**BLAST (4)**|**BLAST (5)**|Monarch|**Monarch (1-3)**|Low-Rank|Dense|
> |---|---|---|---|---|---|---|---|
> |Q/O_proj|1.33706|0.468064|0.324027|0.588342|0.331301|0.107461|0.343395|
> |gate_proj|2.94545|1.20231|0.722091|0.85945|1.48032|0.499424|1.17777|
> |up_proj|4.36134|1.64721|1.05037|2.18552|1.13728|0.676364|1.17128|
> |down_proj|4.1826|1.95465|1.0573|1.97063|0.901401|0.791409|1.30322|
>
> **Llama-3.2-1B Jetson Orin Nano ($n\times B=2048$)**
>
> |Layer|BLAST|**BLAST (4)**|**BLAST (5)**|Monarch|**Monarch (1-3)**|Low-Rank|Dense|
> |---|---|---|---|---|---|---|---|
> |Q/O_proj|10.6432|3.29301|1.42951|3.00342|1.62131|0.613917|1.71007|
> |gate_proj|21.798|7.24673|3.21212|7.31754|3.99097|2.39417|7.57616|
> |up_proj|32.2621|9.78907|4.58255|10.2254|5.27905|3.54127|7.56523|
> |down_proj|32.0027|12.6707|4.85793|8.18524|4.60834|4.26683|9.52961|
>
> **Llama-3.2-1B Jetson Orin Nano ($n\times B=3072$)**
>
> |Layer|BLAST|**BLAST (4)**|**BLAST (5)**|Monarch|**Monarch (1-3)**|Low-Rank|Dense|
> |---|---|---|---|---|---|---|---|
> |Q/O_proj|15.9508|4.1545|1.66329|3.93083|1.95798|0.802238|2.82881|
> |gate_proj|32.6642|10.7296|4.8096|11.2339|6.04505|3.40217|11.4627|
> |up_proj|48.6699|14.5983|6.85099|15.4207|7.86269|5.43419|11.4146|
> |down_proj|48.0443|18.7409|7.15338|12.2364|6.95608|6.34459|14.2694|
>
> **Llama-3.2-1B Jetson Orin Nano ($n\times B=4096$)**
>
> |Layer|BLAST|**BLAST (4)**|**BLAST (5)**|Monarch|**Monarch (1-3)**|Low-Rank|Dense|
> |---|---|---|---|---|---|---|---|
> |Q/O_proj|21.1709|5.5375|2.17735|5.2116|2.58235|1.05405|3.8518|
> |gate_proj|43.4916|14.2908|6.38182|14.6178|7.91519|4.57487|15.2824|
> |up_proj|64.3273|19.5076|9.03011|20.4625|10.2337|7.26835|15.3224|
> |down_proj|64.0272|24.9991|9.41554|16.2874|9.38078|8.54386|19.2533|

---

> ### Author Response · Authors · 2025-12-03
>
> **Q3:** *Baselines. Comparisons appear limited to PyTorch implementations. For a fair performance picture, one could include vendor-optimized libraries, e.g., CUTLASS/cuBLASLt, TensorRT where applicable).*
>
> **A3:** We agree that fair comparisons should rely on optimized baselines. PyTorch implementation baselines already invoke vendor-optimized libraries under the hood, specifically CUTLASS and cuBLAS for `torch.bmm()` and `torch.matmul()`. As a result, the PyTorch implementations we benchmark against are hardware-tuned kernels provided by NVIDIA. This ensures that our comparisons are made against strong, production-grade baselines.
>
> **Q4:** *Portability/generalization. Results focus on Jetson Orin Nano and A40. It is unclear whether the gains transfer to more common data-center GPUs , such as, A100, H100, B100 NVIDIA GPUs. Your evaluation targets Jetson Orin Nano and A40. What performance should readers expect on A100/H100/B100 or similar widely used NVIDIA GPUs? Are there kernel or compiler assumptions that limit portability (e.g., tensor-core MMA variants, shared-memory pressure, register usage, or Triton autotuning constraints)?*
>
> **A4:** Our experiments focus on Jetson Orin Nano and A40 because this is precisely where the bottleneck we study arises. Although global memory is primarily composed of slow DRAM, it also includes a much faster L2 cache implemented in SRAM. When the L2 cache is too small to hold intermediate outputs, these tensors must be repeatedly fetched from DRAM, making kernel boundaries the dominant performance bottleneck. In such settings, fusion becomes essential.
>
> As described in the manuscript, resource-constrained GPUs have L2 caches that are small relative to the intermediate tensors produced by block low-rank layers. For example, A40 (6 MB L2) and Orin Nano (2 MB L2) cannot hold the 32 MB intermediate tensor generated by the first batched matrix multiplication in a BLAST projection for Llama-7B (`bf16`, $16\times1024\times1024$). This mismatch is exactly what motivates our fusion strategy.
>
> In contrast, high-end GPUs such as A100 (40 MB L2) and H100 (128 MB L2) have sufficiently large L2 caches for these intermediates to potentially remain fully on-chip. On such devices, the I/O bottleneck we characterize largely disappears, and the benefits of our technique naturally diminish. For this reason, evaluating on A100/H100 would not be informative for the specific problem our paper targets.
>
> Our scope is intentionally aligned with edge and constrained-GPU deployment, where block low-rank compression is used to fit foundation models on small devices. In these scenarios, if inference becomes slower than dense baselines due to I/O stalls, compression loses much of its practical value. Our experiments therefore focus on the hardware where fusion is most impactful and where these compressed models are actually intended to run.

---

> ### Author Response · Authors · 2025-12-03
>
> **Q5:** *How exactly is the speedup evaluated? For example, [1] in Rule 1 suggests: “When publishing parallel speedup, report if the base case is a single parallel process or best serial execution, as well as the absolute execution performance of the base case. “ Could you report your base case of the dense baseline, and clarify your evaluation according to this suggestion of benchmarking?*
>
> **A5:** Our evaluation protocol follows the benchmarking guidance in Rule 1. The dense implementation serves as the base case, and we report its absolute execution time for both layer-wise and end-to-end inference. Speedups are then computed as the ratio between the dense baseline runtime and the runtime of our method. Here is the dense baseline base-case absolute time for the different model experiments in our work. All times are reported in milliseconds.
>
> **Llama-7B ($n=1024$)**
>
> |Device|Q/K/V/O_proj|gate/up_proj|down_proj|End-to-End|
> |---|---|---|---|---|
> |A40|0.320375|0.850122|0.863710|158.63|
>
> **Llama-3.2-1B ($n=1024$)**
>
> |Device|Q/O_proj|gate_proj|up_proj|down_proj|End-to-End|
> |---|---|---|---|---|---|
> |A40|0.0875804|0.347819|0.348851|0.302584|32.06|
> |Jetson|0.877637|4.35592|3.73001|4.79188|347.9|
>
> **GPT-2 Small ($n=384$)**
>
> |Device|c_attn|c_fc|c_proj|End-to-End|
> |---|---|---|---|---|
> |Jetson|0.407977|0.389531|0.208170|17.87|
>
> **ViT-B ($n=197$)**
>
> |Device|attn_qkv|fc1|fc2|End-to-End|
> |---|---|---|---|---|
> |Jetson|0.277882|0.258549|0.163208|10.52|
>
> **DiT-XL/2 ($n=16384$)**
>
> |Device|QKV_proj|fc1|End-to-End|
> |---|---|---|---|
> |A40|4.47692|1.53223|589.98|
> |Jetson|14.2776|19.2672|7180|
>
> The way runtime is measured is detailed in the appendix. Briefly, for layer-wise measurements we use Triton’s `do_bench()` utility, which repeatedly executes the kernel under controlled conditions and reports averaged latency. End-to-end inference latency is measured using PyTorch’s benchmarking utilities. To avoid cold-start effects (kernel compilation, cache warm-up), we perform several warm-up passes. During measurement, inference runs under `torch.no_grad()` and we call `torch.cuda.synchronize()` to account for CUDA’s asynchronous execution model. Measurements are collected with `torch.utils.benchmark.Timer()`, which executes the forward pass repeatedly for a fixed number of iterations. The resulting averaged runtime is used to compute speedup relative to the dense base case, for both layer-wise and end-to-end evaluations.
>
> **Q6:** *Since BLR can be used for training from scratch, do your findings carry over to training workloads? Can the proposed kernels be used for backward passes, and if so, what speedups versus dense baselines should one expect during pre-training or fine-tuning? If not currently supported, what are the blockers?*
>
> **A6:** Backward propagation corresponds to a different computation pattern than the forward GEMM-like operator, so supporting training would require a separate custom kernel for the backward pass. Whether speedups carry over is not guaranteed: performance would depend on the specifics of the backward formulation, the memory access pattern, and the degree to which fusion opportunities exist. A full backward implementation is therefore outside the scope of this paper.
>
> The scope of this work is inference-time optimization for computing $Y = XW$ where $W$ is a Monarch or BLAST matrix that is trained once and then fixed. This aligns with standard deployment scenarios, where inference dominates cost and runtime, accounting for over 90% of large-scale ML expenditure [4, 5, 6]. Because backpropagation is only needed during training, backward kernels fall outside the scope of this work and are therefore not implemented.
>
> Table 1 reports training results to demonstrate the effectiveness of block low-rank models themselves, independent of our inference-time optimizations. The fused kernels are intended for and evaluated in the inference regime, which is where the performance benefits are most relevant. Extending the fusion strategies to support backpropagation is an interesting direction for future work.
>
> [4] HPCwire. AWS Upgrades its GPU-Backed AI Inference Platform, 2019. https://www.hpcwire.com/aiwire/2019/03/19/aws-upgrades-its-gpu-backed-ai-inference-platform/
>
> [5] Barr, J. Amazon EC2 Update – Inf1 Instances with AWS Inferentia Chips for High Performance Cost-Effective Inferencing, 2019. https://aws.amazon.com/blogs/aws/amazon-ec2-update-inf1-instances-with-aws-inferentia-chips-for-high-performance-cost-effective-inferencing/
>
> [6] Gonon, A., Zheng, L., Carrivain, P., and Le, Q.T. Fast Inference with Kronecker-Sparse Matrices. In ICML, 2025.

---

### Official Review · Reviewer_cefX · 2025-10-30

**Soundness:** 3
**Presentation:** 2
**Contribution:** 1
**Rating:** 2
**Confidence:** 4

**Summary:**

This paper identifies costly data movement operations when computing on GPU the matrix multiplication associated with structured matrices such as Block low rank (related to Monarch matrices) and BLAST matrices, in the context of inference with compressed foundation models (vision transformers, large language models, diffusion transformers). They tackle this issue by introducing kernel fusion to reduce the cost of these memory operations. Experiments show both per-layer and end-to-end speedup at inference compared to previous implementations of matrix multiplication with such structured matrices.

**Strengths:**

* The paper evaluates the method on the compression of a broad range of foundation models, demonstrating the general applicability of the proposed method. Reported speedups are obtained under realistic conditions, such as BF16 precision and multi-token inference, which enhances the practical relevance of the experiments.
* The speedup analysis is conducted at both the layer and end-to-end levels, providing insights into the sources of efficiency gains. The trade-off between accuracy and inference speed is clearly presented in Table 1 and Figures 9 and 10, offering a valuable perspective on the method’s strengths and limitations relative to prior approaches, including low-rank and dense baselines.

**Weaknesses:**

My main concern with this submission is its positioning with respect to a very close paper [1] that was not mentioned in the submission. [1] also proposes to reduce the cost of data movement operations when performing matrix multiplication on GPU with the so-called Kronecker-sparse factors. These matrices are typically involved in butterfly factorizations and in the Monarch matrices discussed in this submission. In [1], it is observed that the original implementation of Monarch matrix multiplication [2] has costly memory operations due to permutations (as claimed in the authors' submission). Then, [1] proposes a novel tiling that avoids the costs of these permutations, and implements it with a CUDA kernel to show effective speedup compared to previous baselines. This overall narrative appears highly similar to that of the present paper.

I highly recommend the authors to clarify their positioning with resepct to [1]. What are the similarities, differences and novelties compared to [1]?

[1] Gonon, A., Zheng, L., Carrivain, P., and Le, Q.T. Fast Inference with Kronecker-Sparse Matrices. In ICML, 2025.

[2] Dao, T., Chen, B., Sohoni, N. S., Desai, A. D., Poli, M., Grogan, J., Liu, A., Rao, A., Rudra, A., and Ré, C. Monarch: Expressive structured matrices for efficient and accurate training. In ICML, 2022.

**Questions:**

* What are the expected benefits of implementing the kernel using Triton vs. CUDA in this specific setting?
* Does the proposed kernel implementation leverage tensorcores?
* The experiments were carried on BF16, does the authors expect a speedup compared to baselines when using FP16 instead?
* How much effort was devoted to hyperparameter tuning in Table 1 when comparing the accuracy of the various compressed foundation models? The significance of Table 1 may depend on the quality of the benchmarking protocol, since it is important to ensure that all methods are compared on equal footing, with proper tuning on a separate validation set.

---

> ### Author Response · Authors · 2025-12-03
>
> We thank the reviewer for the feedback and remarks. We address each of the reviewer’s comments and questions in the following.
>
> **Q1:** *My main concern with this submission is its positioning with respect to a very close paper [1] that was not mentioned in the submission. [1] also proposes to reduce the cost of data movement operations when performing matrix multiplication on GPU with the so-called Kronecker-sparse factors. These matrices are typically involved in butterfly factorizations and in the Monarch matrices discussed in this submission. In [1], it is observed that the original implementation of Monarch matrix multiplication [2] has costly memory operations due to permutations (as claimed in the authors' submission). Then, [1] proposes a novel tiling that avoids the costs of these permutations, and implements it with a CUDA kernel to show effective speedup compared to previous baselines. This overall narrative appears highly similar to that of the present paper. I highly recommend the authors to clarify their positioning with resepct to [1]. What are the similarities, differences and novelties compared to [1]?*
>
> **A1:** Thank you for bringing this concurrent work to our attention. The ICML25 paper [1] also observes that Monarch-style factorizations incur high data-movement overhead due to permutations, and proposes a CUDA kernel with a custom tiling strategy to mitigate this. However, the setting and scope of [1] differ substantially from ours.
>
> First, their implementation targets FP32 inference on an A100-PCIe-40GB GPU, without tensor cores and using a larger-than-necessary precision, which limits relevance for contemporary deployment where lower-precision formats (`bf16`, `fp16`, `int8`) dominate. Their reported speedups are modest (median $\approx1.4\times$) and end-to-end results are provided only for ViT-S and GPT-2 Medium. In contrast, our work evaluates a broader and more modern set of models, including Llama-3.2-1B, across vision, language, and diffusion.
>
> Second, [1] implements kernels highly specialized for a single ViT batch size (25088) and ~700 patterns, and the authors explicitly note that support for other batch sizes, patterns, and lower-precision formats remains future work. By comparison, our Triton kernels are portable across data types, GPUs, and model/layer configurations.
>
> Finally, [1] does not use `torch.compile()` for baselines, which we show provides substantial baseline improvements, changing both the absolute and relative performance landscape. Our contributions are not simply fusing permutations, but identifying where fusion is helpful, restructuring the block low-rank dataflow, and introducing operation-reordering strategies. These techniques could be incorporated into future CUDA or Triton backends. We plan to discuss [1] in a revised version of our manuscript upon acceptance.
>
> **Q2:** *What are the expected benefits of implementing the kernel using Triton vs. CUDA in this specific setting?*
>
> **A2:** Triton provides several advantages in this setting. Most importantly, it offers portability across precisions (`bf16`, `fp16`, `int8`), architectures, and compiler backends with far less hand-tuning than a CUDA implementation would require. Because Monarch/BLAST use reshapes, permutations, and batched low-rank matmuls whose legality and optimal ordering depend on tensor shapes, a Triton implementation allows us to rapidly adapt kernels to different model sizes, block patterns, and hardware constraints, something that would require substantial re-engineering in CUDA.
>
> Additionally, Triton integrates naturally with compiler-driven optimization pipelines such as `torch.compile()`, enabling our kernels to coexist with automatic fusion and scheduling passes. This is essential for deployment on small-memory edge GPUs, where portability and maintainability matter as much as peak performance. Triton enables us to implement our fusion and data-layout strategies in a way that is both efficient and broadly applicable. A CUDA approach would sacrifice generality for hardware-specific tuning.

---

> ### Author Response · Authors · 2025-12-03
>
> **Q3:** *Does the proposed kernel implementation leverage tensorcores?*
>
> **A3:** Yes. With the exception of the second batched matmul in BLAST (4) as noted in the paper, all other fused kernels in our implementation make full use of tensor cores.
>
> **Q4:** *The experiments were carried on BF16, does the authors expect a speedup compared to baselines when using FP16 instead?*
>
> **A4:** ​​Ampere GPUs provide tensor-core support for both `bf16` and `fp16`, so we expect similar speedups when switching to `fp16`. Converting our kernels from `bf16` to `fp16` requires only minor changes in Triton (e.g., relaxing datatype assertions and adjusting final casts), and the `fp32` to `fp16` cast is a negligible fraction of the overall runtime. One benefit of using Triton is this portability: the fused kernels generalize cleanly across `bf16` and `fp16` with minimal modification, so performance gains could carry over.
>
> **Q5:** *How much effort was devoted to hyperparameter tuning in Table 1 when comparing the accuracy of the various compressed foundation models? The significance of Table 1 may depend on the quality of the benchmarking protocol, since it is important to ensure that all methods are compared on equal footing, with proper tuning on a separate validation set.*
>
> **A5:** The accuracy results in Table 1 rely on results and procedures established in prior work, which has been published at top-tier ML venues (ICML, NeurIPS). These works performed controlled comparisons with proper hyperparameter tuning and validation for different compression methods, and we follow the same methodology.
>
> For Llama-3.2-1B, which is the only additional model we introduce beyond prior work, we adopt the same tuning and validation protocol described in those papers to ensure comparability and fairness. Details about the training and validation setup used for this experiment are also provided in the appendix.

---

### Official Review · Reviewer_ZYGz · 2025-10-31

**Soundness:** 2
**Presentation:** 3
**Contribution:** 2
**Rating:** 2
**Confidence:** 4

**Summary:**

This paper studies the GPU kernel performance problem of block low-rank (BLR) compression for linear layers. It utilizes the roofline model to identify that the state-of-the-art BLR computations are memory bounded. It proposes two sets of fused Triton kernels for Monarch and BLAST algorithms and achieves good performance improvement than the baseline.

**Strengths:**

1. It explores the rarely studied problem of optimizing the GPU kernels of block low-rank compression.
2. It presents a set of practical solutions for the problem.

**Weaknesses:**

1. Given that the low-rank compression's accuracy is bad, it lacks a well discussed motivation of optimizing this kind of algorithm specifically.
2. The roofline model analysis is not novel nor necessary. The performance bottleneck of the BLR computation is obvious from system aspect, that the matmul shape is quite small so that it becomes memory bounded (especially the intermediate K dimension of the matmul).
3. The GPU kernel optimization is straight forward and does not have new contribution from the system aspect.
4. It has wrong FLOP estimation of all the matmul problems in this paper.

**Questions:**

1. From Table.1, when CF = 2x, the accuracy of BLR seems pretty bad. Specifically, if the CF = 2x for quantization (i.e., quantizing from float16 to 8-bit), the performance will be nearly lossless. It would be better to have a discussion of the advantages of using low-rank compression, especially when compared to quantization.
2. It is well known that small matmul shape will lead to low compute efficiency, either parallelism problem when M/N is small, or software pipeline problem when K is small (assuming a MNK matmul problem). It seems the roofline analysis is not a new contribution.
3. The CUTLASS library has rich examples of fused permutation and back-to-back matmul. The methodologies in this manuscript does not have new insight for the operation fusion problem.
4. For a matmul of MNK problem (i.e., [M,K] * [K,N]), the FLOP is 2MNK including both the multiplication and accumulation instructions. This paper wrongly estimates it as MNK in Section 2.
5. The model used in this paper is old.
6. In Figure 3, why the low-rank version has a larger speedup when the batchsize is larger? As for batch size 1, it has reduced half of the memory traffic, but only achieves 1.5x speedup. Have you analyzed why?

---

> ### Author Response · Authors · 2025-12-03
>
> We thank the reviewer for the feedback and remarks. We address each of the reviewer’s comments and questions in the following.
>
> **Q1:** *Given that the low-rank compression's accuracy is bad, it lacks a well discussed motivation of optimizing this kind of algorithm specifically. From Table.1, when CF = 2x, the accuracy of BLR seems pretty bad. Specifically, if the CF = 2x for quantization (i.e., quantizing from float16 to 8-bit), the performance will be nearly lossless. It would be better to have a discussion of the advantages of using low-rank compression, especially when compared to quantization.*
>
> **A1:** Block low-rank compression and quantization are orthogonal. In practice, both techniques can be combined for additional gains [1]. Our work does not preclude applying quantization on top of block low-rank layers.
>
> Regarding accuracy, the results in Table 1 show that when the model size is moderate (e.g., ViT-B, GPT2-S), block low-rank methods can recover nearly all accuracy. For larger models, we agree that vanilla low-rank can degrade accuracy substantially. This is precisely why recent methods such as Monarch and BLAST were introduced, to overcome the limitations of simple low-rank factorization. As shown in prior work [2], Monarch and BLAST recover accuracy significantly faster and more reliably than standard low-rank approaches, even when trained for only a fraction of an epoch on a reduced dataset (SlimPajama). Our results in this paper follow the same pattern: the more expressive block-structured variants achieve better recovery than pure low-rank, particularly at moderate compression factors.
>
> Regarding the generality and far-reaching impact of block low-rank matrix multiplications, block low-rank families have strong potential to be universal components not only in deep learning but also in machine learning and signal processing. Improving the memory efficiency of Monarch and BLAST matrix multiplications, as done in this work, will further encourage their adoption.
> From the deep learning standpoint, [3] found that transformers with Monarch or Block Tensor-Train (BTT) weight matrices follow better scaling laws than networks with dense weight matrices under the same FLOP budget. BLAST also follows the conditions identified by [3] for outperforming the scaling laws of dense models. This suggests that networks with the block low-rank families (Monarch, BTT, and BLAST) can serve as standard components in deep learning, as they can deliver higher accuracy given a fixed training budget.
>
> From the machine learning (ML) and signal processing (SP) perspective, Monarch and BLAST matrices can provide a GPU-friendly implementation of various transforms prevalent in SP and ML in general, including discrete Fourier transforms (DFT), and discrete sine/cosine transforms (DST/DCT) [4]. Since many ML/SP research and applications rely on GPUs, Monarch and BLAST can be adopted to utilize GPU resources efficiently. For example, [5]  implemented a DFT using Monarch matrices to leverage tensor cores on NVIDIA GPUs for higher throughput.
>
> We also emphasize that this paper substantially improves the memory efficiency of Monarch and BLAST matrix multiplication routines, which will further support their adoption in the ML, deep learning, and SP communities.
>
> [1] Saha et al. "Compressing large language models using low rank and low precision decomposition." NeurIPS24.
>
> [2] Lee et al. “BLAST: Block-Level Adaptive Structured Matrices for Efficient Deep Neural Network Inference.” NeurIPS24.
>
> [3] Qiu et al. “Compute Better Spent: Replacing Dense Layers with Structured Matrices.” ICML24.
>
> [4] Dao et al. “Monarch: Expressive Structured Matrices for Efficient and Accurate Training.” ICML22.
>
> [5] Fu et al. "FlashFFTConv: Efficient Convolutions for Long Sequences with Tensor Cores." ICLR24.

---

> ### Author Response · Authors · 2025-12-03
>
> **Q2:** *The roofline model analysis is not novel nor necessary. The performance bottleneck of the BLR computation is obvious from system aspect, that the matmul shape is quite small so that it becomes memory bounded (especially the intermediate K dimension of the matmul). It is well known that small matmul shape will lead to low compute efficiency, either parallelism problem when M/N is small, or software pipeline problem when K is small (assuming a MNK matmul problem). It seems the roofline analysis is not a new contribution.*
>
> **A2:** It is true that low-rank compression reduces the intermediate dimension. However, in our setting the resulting inner dimension, $r$, remains quite large (up to 1536; see Table 3), and the batched structure of the computation (with up to 16 blocks per layer) preserves substantial parallelism. For this reason, the performance is not fully explained by the standard “small matmul = low compute efficiency” argument.
>
> Our goal with the roofline analysis as stated in our manuscript is not to claim novelty, but to provide intuition for the empirical bottlenecks and motivate our optimizations. The analysis clarifies why BLR matmuls become memory-bound at realistic configurations and directly informs our choices of partial fusion, memory-layout improvements, and operation reordering to increase effective arithmetic intensity.
>
> **Q3:** *The GPU kernel optimization is straightforward and does not have new contributions from the system aspect. The CUTLASS library has rich examples of fused permutation and back-to-back matmul. The methodologies in this manuscript does not have new insight for the operation fusion problem.*
>
> **A3:** Our contribution is not simply performing fusion, but identifying where fusion is feasible, which operations can be legally combined, and how to restructure the block low-rank dataflow so that reshapes/permutations and batched low-rank matrix multiplications execute faster than their dense counterparts. These patterns have not been previously explored in CUTLASS or its examples, which PyTorch can already target under the hood when using CUDA for matrix multiplications.
>
> We chose Triton to ensure portability and compatibility with modern compiler-based optimization pipelines. Triton currently cannot automatically fuse the reshapes and permutations required by Monarch/BLAST with the subsequent batched matrix multiplications. Our work provides a concrete implementation that demonstrates such fusion is possible and yields significant speedups on resource-constrained GPUs. These results also provide actionable insights for compiler designers and can inform future Triton or PyTorch fusion strategies.
>
> **Q4:** *It has wrong FLOP estimation of all the matmul problems in this paper. For a matmul of MNK problem (i.e., [M,K] * [K,N]), the FLOP is 2MNK including both the multiplication and accumulation instructions. This paper wrongly estimates it as MNK in Section 2.*
>
> **A4:** Thank you for pointing out the typo in the FLOP calculation in Section 2. We have updated the table to reflect the correct $2\times M\times N\times K$ formulation. Importantly, all reported speedups in the paper are based on empirical measurements, not FLOP estimates, so the results and conclusions are unaffected by this typographical error.
>
> **Q5:** *The model used in this paper is old.*
>
> **A5:** The models in this paper were chosen to be diverse and representative across a range of tasks, including vision, language, and image generation. In contrast to [6] which is concurrent work on a similar problem using highly-sparse Kronecker matrices which evaluated a smaller subset of models (ViT-S, GPT2-M), our evaluation includes additional and more recent models, such as Llama-3.2-1B, to better reflect contemporary architectures.
>
> [6] Gonon, A., Zheng, L., Carrivain, P., and Le, Q.T. Fast Inference with Kronecker-Sparse Matrices. In ICML, 2025.

---

> ### Author Response · Authors · 2025-12-03
>
> **Q6:** *In Figure 3, why the low-rank version has a larger speedup when the batchsize is larger? As for batch size 1, it has reduced half of the memory traffic, but only achieves 1.5x speedup. Have you analyzed why?*
>
> **A6:** The speedups in Figure 3 were measured on an A40 GPU, so we can only partially attribute the exact behavior, but two factors could help explain the difference. First, the gap between single-token and multi-token speedups is actually modest: the single-token speedups are {$1.60$, $1.71$, $1.65$}$\times$, while the multi-token speedups are {$1.70$, $1.85$, $1.85$}$\times$. So the effect is smaller than it may initially appear.
>
> Second, during single-token inference the overall runtime is much shorter (~$5\times$ smaller than multi-token), which makes fixed overheads, especially kernel-launch latency, account for a larger fraction of end-to-end time. Because the low-rank path introduces an additional kernel launch relative to the dense baseline, this overhead disproportionately impacts the single-token regime, reducing the effective speedup. At larger sequence lengths, this overhead is amortized across more computation, allowing the reduction in actual compute work to dominate and resulting in the slightly higher observed speedups.

---

### Official Review · Reviewer_c6Dq · 2025-11-01

**Soundness:** 3
**Presentation:** 3
**Contribution:** 2
**Rating:** 2
**Confidence:** 4

**Summary:**

This paper presents a memory-efficient acceleration framework for large transformer-based foundation models using block low-rank (BLR) compression techniques such as Monarch and BLAST. While BLR methods theoretically reduce FLOPs and model size, the authors show via roofline analysis that multi-token inference on GPUs like the Jetson Orin Nano and A40 often becomes memory-bound, limiting practical speedups. To address this, they design custom Triton kernels featuring partial fusion, operation reordering, and optimized memory layouts to minimize data movement overhead.

**Strengths:**

1. Demonstrating up to 3.76× speedup and 3× model size reduction on resource-constrained GPUs (Jetson Orin Nano, A40) highlights real-world relevance for edge and low-memory environments.
2. The kernel fusion method is clearly explained.

**Weaknesses:**

1. The main issue lies in the limited novelty of the contribution. Using kernel fusion to reduce I/O overhead is a well-established engineering practice in modern LLM systems. State-of-the-art training frameworks such as Megatron-LM already include numerous fused operations that consistently outperform vanilla PyTorch implementations. Therefore, the proposed optimization should be viewed primarily as an engineering refinement rather than an academic innovation. For comparison, methods like FlashAttention [1] introduce genuine algorithmic innovations (e.g., the online softmax formulation). Moreover, FlashAttention targets the self-attention mechanism, which is a core and widely used component across almost all transformer architectures, making its impact general and far-reaching. In contrast, this paper focuses on specialized optimizations for Monarch and BLAST, which are relatively niche techniques within block low-rank modeling.

2. Even as an engineering contribution, several concerns remain:

    2.1 Code availability – the implementation is not released.

    2.2 Backward support – it is unclear whether the custom kernels include backward computation. Moreover, the training results in Table 1 seems do not use the fused kernels, which raises questions about their applicability during training.

    2.3 Experimental scope – all experiments are performed on low-memory GPUs (e.g., Jetson Orin Nano, A40). Evaluations on modern high-end GPUs such as A100 or H100 are necessary to demonstrate scalability and broader relevance.

[1] Dao, Tri, et al. "Flashattention: Fast and memory-efficient exact attention with io-awareness." Advances in neural information processing systems 35 (2022): 16344-16359.

**Questions:**

Check above.

---

> ### Author Response · Authors · 2025-12-03
>
> We thank the reviewer for the feedback and remarks. We address each of the reviewer’s comments and questions in the following.
>
> **Q1:** *The main issue lies in the limited novelty of the contribution. Using kernel fusion to reduce I/O overhead is a well-established engineering practice in modern LLM systems such as in FlashAttention [1].*
>
> **A1:** We agree that kernel fusion as a general technique for reducing I/O overhead is well established, and we do not claim novelty in the idea of fusion itself. Our contribution is in demonstrating that existing fusion strategies, such as those used in FlashAttention or Megatron-LM, do not directly apply to compressed linear layers using block low-rank matrix multiplication (e.g., Monarch, BLAST).
> FlashAttention is able to perform full fusion largely because attention heads are independent and each head dimension is small ($\leq128$). This structure makes it possible to fuse the entire attention pipeline. We allude to this in Section 4 and Appendix A.1, where we talk about the applicability of full fusion in traditional low-rank matrix multiplication for small ranks ($\leq128$). In contrast, block low-rank layers do not share these properties. Their factorized representations prevent full fusion, and the optimal boundaries for partial fusion are non-obvious. To our knowledge, prior work has not studied where fusion is feasible or effective in block low-rank operators, nor how to fuse reshapes, permutations, and batched low-rank matrix multiplication operations while preserving correctness and achieving speedups.
>
> Furthermore, current compilation approaches (e.g., Triton) are unable to automatically fuse reshapes with batched matrix multiplications in these settings. Our work provides a concrete design showing how these operations can be combined into fewer kernels, enabling consistent performance gains on resource-constrained GPUs such as Jetson Orin Nano and A40. In this sense, the novelty lies not in the use of fusion, but in extending fusion methodologies to a setting where they have not been previously explored or supported by compilers.
>
> [1] Dao, Tri, et al. "Flashattention: Fast and memory-efficient exact attention with IO-awareness." Advances in neural information processing systems 35 (2022): 16344-16359.
>
> **Q2:** *FlashAttention [1] targets the self-attention mechanism, which is a core and widely used component across almost all transformer architectures, making its impact general and far-reaching. In contrast, this paper focuses on specialized optimizations for Monarch and BLAST, which are relatively niche techniques within block low-rank modeling.*
>
> **A2:** We understand the reviewer’s concern. However, the block low-rank families have strong potential to be universal components not only in deep learning but also in machine learning and signal processing. Improving the memory efficiency of Monarch and BLAST matrix multiplications, as done in this work, will further encourage their adoption.
>
> From the deep learning standpoint, [2] found that transformers with Monarch or Block Tensor-Train (BTT) weight matrices follow better scaling laws than networks with dense weight matrices under the same FLOP budget. BLAST also follows the conditions identified by [2] for outperforming the scaling laws of dense models. This suggests that networks with the block low-rank families (Monarch, BTT, and BLAST) can serve as standard components in deep learning, as they can deliver higher accuracy given a fixed training budget.
>
> From the machine learning (ML) and signal processing (SP) perspective, Monarch and BLAST matrices can provide a GPU-friendly implementation of various transforms prevalent in SP and ML in general, including discrete Fourier transforms (DFT), and discrete sine/cosine transforms (DST/DCT) [3]. Since many ML/SP research and applications rely on GPUs, Monarch and BLAST can be adopted to utilize GPU resources efficiently. For example, [4]  implemented a DFT using Monarch matrices to leverage tensor cores on NVIDIA GPUs for higher throughput.
>
> We also emphasize that this paper substantially improves the memory efficiency of Monarch and BLAST matrix multiplication routines, which will further support their adoption in the ML, deep learning, and signal processing communities.
>
> [2] Qiu et al. “Compute Better Spent: Replacing Dense Layers with Structured Matrices.” ICML24.
>
> [3] Dao et al. “Monarch: Expressive Structured Matrices for Efficient and Accurate Training.” ICML22.
>
> [4] Fu et al. "FlashFFTConv: Efficient Convolutions for Long Sequences with Tensor Cores." ICLR24.

---

> ### Author Response · Authors · 2025-12-03
>
> **Q3:** *The implementation is not released.*
>
> **A3:** The full implementation of all methods described in the paper was included in the supplementary material section of OpenReview at submission time. As indicated in the manuscript, we will also make an official, public release of the code upon acceptance, consistent with standard ICLR practice. If any component was difficult to locate in the supplementary package, we are happy to clarify, but all implementation details were provided.
>
> **Q4:** *It is unclear whether the custom kernels include backward computation. Moreover, the training results in Table 1 do not use the fused kernels, which raises questions about their applicability during training.*
>
> **A4:** As stated in the paper, the scope of this work is inference-time optimization for computing $Y = XW$ where $W$ is a Monarch or BLAST matrix that is trained once and then fixed. This aligns with standard deployment scenarios, where inference dominates cost and runtime, accounting for over 90% of large-scale ML expenditure [4, 5, 6]. Because backpropagation is only needed during training, backward kernels fall outside the scope of this work and are therefore not implemented.
>
> Regarding Table 1, the table reports training results to demonstrate the effectiveness of block low-rank models themselves, independent of our inference-time optimizations. The fused kernels are intended for and evaluated in the inference regime, which is where the performance benefits are most relevant. Extending the fusion strategies to support backpropagation is an interesting direction for future work.
>
> [4] HPCwire. AWS Upgrades its GPU-Backed AI Inference Platform, 2019. https://www.hpcwire.com/aiwire/2019/03/19/aws-upgrades-its-gpu-backed-ai-inference-platform/
>
> [5] Barr, J. Amazon EC2 Update – Inf1 Instances with AWS Inferentia Chips for High Performance Cost-Effective Inferencing, 2019. https://aws.amazon.com/blogs/aws/amazon-ec2-update-inf1-instances-with-aws-inferentia-chips-for-high-performance-cost-effective-inferencing/
>
> [6] Gonon, A., Zheng, L., Carrivain, P., and Le, Q.T. Fast Inference with Kronecker-Sparse Matrices. In ICML, 2025.
>
> **Q5:** *All experiments are performed on low-memory GPUs (e.g., Jetson Orin Nano, A40). Evaluations on modern high-end GPUs such as A100 or H100 are necessary to demonstrate scalability and broader relevance.*
>
> **A5:** Our experiments focus on Jetson Orin Nano and A40 because this is precisely where the bottleneck we study arises. Although global memory is primarily composed of slow DRAM, it also includes a much faster L2 cache implemented in SRAM. When the L2 cache is too small to hold intermediate outputs, these tensors must be repeatedly fetched from DRAM, making kernel boundaries the dominant performance bottleneck. In such settings, fusion becomes essential.
>
> As described in the manuscript, resource-constrained GPUs have L2 caches that are small relative to the intermediate tensors produced by block low-rank layers. For example, A40 (6 MB L2) and Orin Nano (2 MB L2) cannot hold the 32 MB intermediate tensor generated by the first batched matrix multiplication in a BLAST projection for Llama-7B (`bf16`, $16\times1024\times1024$). This mismatch is exactly what motivates our fusion strategy.
>
> In contrast, high-end GPUs such as A100 (40 MB L2) and H100 (128 MB L2) have sufficiently large L2 caches for these intermediates to potentially remain fully on-chip. On such devices, the I/O bottleneck we characterize largely disappears, and the benefits of our technique naturally diminish. For this reason, evaluating on A100/H100 would not be informative for the specific problem our paper targets.
>
> Our scope is intentionally aligned with edge and constrained-GPU deployment, where block low-rank compression is used to fit foundation models on small devices. In these scenarios, if inference becomes slower than dense baselines due to I/O stalls, compression loses much of its practical value. Our experiments therefore focus on the hardware where fusion is most impactful and where these compressed models are actually intended to run.

---

### Note · Program_Chairs · 2026-01-17
**Submission Desk Rejected by Program Chairs**

The following references in this submission do not refer to real documents and/or have major errors in bibliographic information:

 Chi Yang, Sara Seif Baghsorkhi, Karthik Muralidharan, and John Cavazos. An empirical roofline methodology for gpus: Analyzing performance portability. In Proceedings of the ACM International Conference on Computing Frontiers, pp. 1–10. ACM, 2013. doi: 10.1145/2482767. 2482798.